# Effect of Defects Part I: Degradation of Constitutive Coefficients as an Input to the Composite Failure Model with Microvoids and Porosity

**Vahid Tavaf and Sourav Banerjee ***

College of Engineering and Computing, University of South Carolina, Columbia, SC 29208, USA;
vahid.tavaf@gmail.com
* Correspondence: banerjes@cec.sc.edu

**Abstract:** It is always challenging to provide appropriate material properties for a composite progressive failure model. The nonstandard percentage reduction method that is commonly used to degrade the material constants with micro-scale defects generates tremendous uncertainty in failure prediction. The constitutive matrix is composed of multiple material constants. It is not necessary that all constants degrade either equally or linearly due to a certain state of material defects. With this very concern in mind, this article presents a guideline for using a quantified perturbation for each coefficient appropriately. It also presents distribution of effective material properties (EMPs) in unidirectional composite materials with different states of defects such as voids. Irrespective of resin transfer molding (RTM) or chemical vapor infiltration (CVI) processes, manufacturers' defects such as voids of different shapes and sizes are the most common that occur in composite materials. Hence, it is important to quantify the 'effects of defects' void content herein on each material coefficient and EMP. In this article, stochastically distributed void parameters such as the void content by percent, size, shape, and location are considered. Void diameters and shapes were extracted from scanning acoustic microscope (SAM) images of 300,000 cycles of a fatigued composite. The EMPs were calculated by considering unit cells, homogenization techniques, and micromechanical concepts. The periodic boundary conditions were applied to unit cells to calculate the EMPs. The result showed that EMPs were degraded even when there was a small percentage of the void content. More importantly, the constitutive coefficients did not degrade equally but had a definitive pattern.

**Keywords:** effect of defects; progressive failure model; progressive failure analysis (PFA); microvoids; porosity; composite model; composite failure model; constitutive coefficients; composite material properties; material property; RTM; CVI; thermoplastic; thermoset; composites

## 1. Introduction

Composite materials are extensively used in a wide range of industries including aerospace, automotive, marine, civil infrastructure, and biomedical applications. Failure of composite materials starts with distributed defects or local degradation, which is known as early-stage damage. The initial damages may significantly influence the local material properties of composite materials due to local compliance [1]. Local degradation of material properties leads to damage evolution in a unique way and ultimately causes the failure of the structures [2]. This demands an accurate understanding of the 'effects of defects'. Such effects start from the micro-scale and should be accounted for quantitatively. Hence, accurate and more effective input of local material properties to the progressive composite failure models (e.g., at the Gauss points of a finite element) is required. To determine the actual strength of the structures using progressive failure models, the correct material properties are required [3]. Therefore, the quantification of material properties with local degradation is very important. In addition, as a byproduct, the degraded material properties could be used in computational nondestructive evaluations (CNDE). To overcome

the limitations of traditional structural health monitoring (SHM) and NDE and to better understand the sensor signals for accurate and confident detection of initial damage at an early stage, degraded material properties are required; however, this is currently impossible at this state of the art [4].

Generally, four types of early-stage damages occur in composite materials. (a) manufacturing of components/fibers/prepregs, (b) Composite manufacturing, (c) Installation, and (d) Exploitation/ageing and recycling [5]. The first two items occur when an intended design process deviates during manufacturing. For example, the fiber and resin may be separated before or during manufacturing. The third and fourth items usually take place in composites during operation due to local failures. Manufacturing defects are inevitable in all composite materials due to resin transfer molding (RTM) or chemical vapor infiltration (CVI) processes [6]. During the operation of in-service composites, the early-stage defects evolve during the first 30% of their lifespan [7,8]. Microvoids are the most common manufacturing and in-service defects that occur in composite materials. During the manufacturing of fiber-reinforced composites using the RTM process, the resin is injected into the fiber tow preforms. As a result of air trapped within the fiber and matrix during injection molding, microvoids are formed. They are easy to eliminate by the viscous flow of the resin [9]. The location, shape, and size of the microvoids can be random. They contribute to degraded local stiffness and consequently cause uncertain local effective material properties (EMPs) compared to the target design. Effects of micro defects, i.e., the effect of microvoids on the EMPs are addressed in this article.

Researchers have conducted studies on the effect of voids on different properties of composite materials such as strength and elastic properties [10–12]. In studying progressive damage modeling, Barsoum and Faleskog [10] performed a micromechanical analysis to determine the influence of load parameters on void growth and coalescence. Jacques et al. [11] conducted a study on progressive damage modeling into ductile solid materials due to void nucleation and void growth. The void nucleation caused faster damage accumulation and earlier occurrence of fracture. Hyde et al. [12] studied the effect of micro-voids on the strength of unidirectional (UD) fiber-reinforcement composite materials using the micromechanics-based finite element method. They considered two different void models. They distributed standard spherical void shapes in the matrix and a single inter-fiber void with triangular, circular, square, and pentagonal shapes. They found that the pentagonal inter-fiber void type had a greater effect on the strength of the composite materials, which decreased by a higher percentage than the other void shapes in the composite.

Porous materials can be considered the limiting case of composites in which the third phase is the pores [13]. Additionally, voids are also considered to contribute to porosity and can be introduced as a third component in fiber-reinforced plastics. Therefore, for years, the quantification of the elastic properties of porous media has been a topic of interest. Li and et al. [14] presented a 2D model to calculate the elastic properties of randomly distributed void models for porous materials using the finite element method. The results confirmed that the elastic properties of porous materials were not sensitive to the shape of the voids. Khadi and et al. [15] studied the effect of randomly distributed and randomly oriented voids with different volume fractions and shapes on the overall yield surface of porous media using computational micromechanics. The results showed that the overall yield surface was independent of the void shapes, and this was later confirmed by Tavaf et al. [1]. Chao et al. [16] studied the effect of porosity on the flexural properties of 2D carbon/carbon composites. The results showed that an increase in porosity may accelerate the damage in 90° plies and delamination may be aggravated. Qui and et al. [17] calculated the transverse mechanical properties of UD composites with irregular pores. They found that the effective transverse elastic properties decreased when the porosity and the pores of clustering increased. Chung and et al. [18] evaluated the effect of a cluster of voids with different void sizes on the thermal and mechanical properties of concrete. They concluded that when the size of voids increases, the directional modulus decreases. Please note that most studies referred to above reported the effect of defects on the engineering coefficients, e.g.,

Young's modulus only. However, the effect of defects on different constitutive coefficients may be different and cannot be easily retrieved from the degraded Young's modulus. As the degradation is nonlinear, we argue that providing degraded constitutive (i.e., separate degradation of different coefficients in the constitutive matrix) coefficients is more effective than providing a degraded engineering coefficient.

Lambert et al. [19] introduced a model with a large individual void in a laminated matrix. They reported that the largest void had a significant effect on fatigue life. The effect of voids on the elastic properties and strength of composite materials has been studied for decades. Huang and Talerja [20] presented the effect of a micro-void with a cylindrical and elliptical cross-sectional area on the elastic properties of UD fiber-reinforced composite. The results showed that the void content contributed significantly to the degradation of the out-of-plane modulus, while the in-plane properties had minimal effect. Swaminathan et al. [21] presented a two-dimensional RVE model with nonuniform dispersion of fibers for a UD composite to calculate the EMPs. The results showed that the size of RVE would be 50 μm when the fiber volume fraction in a pristine state was between 31 and 33%. Additionally, they proposed an appropriate size of an RVE for UD composites undergoing interfacial debonding. They found that the size of RVE with a 32% fiber volume fraction should be 63 μm [22].

Protz et al. [23] investigated the effect of voids on the rate-dependent material properties and fatigue behavior of non-crimp fabric composite materials. Choudhry et al. [24] modeled 8-harness satin weave glass fiber-reinforced composites using a micromechanics-based finite element method. They predicted the material properties of the composite material in the pristine state and in the presence of voids. Chang et al. [25] evaluated the mechanical behavior of UD fiber-reinforced composites with different void morphologies using the finite element method. They verified the results with the Halpin–Tsai equation and experimental characterization from the literature. They found that the voids with a higher aspect ratio (width to height) had less degradation on the in-plane modulus but a larger reduction of the out-of-plane modulus. Huang and Gong [6] predicted the elastic properties of a 3D woven composite in the presence of voids using microscale RVE containing void components. The results showed that the elastic properties decreased due to the void content. Additionally, the results obtained were in good agreement with the analytical results. Gohari and et al. [26] presented an approach toward localized failure inspection of pressurized ellipsoidal domes made with a woven composite. The results showed that the circumferential regions near the meridian in prolate and oblate ellipsoidal domes tolerated the highest deformation under internal pressure. Ekoi and et al. [27] evaluated the mechanical properties (tensile, flexural, and fatigue) of a woven, continuous carbon fiber composite printed by additive manufacturing. They compared the material properties of the woven composite to nonwoven AM printed composites (unidirectional and multidirectional fibers). The tensile strength of the woven composites was 52% lower than that attained by the unidirectional (nonwoven) fiber composites and 38% higher than the multidirectional (nonwoven) fiber composites.

Hyde and Lui [28] calculated the strength of UD-reinforced composite materials with fiber waviness and void defects. Gao and et al. [29] calculated the elastic properties of braided composites with void defects. They also predicted the strength of the braided composite with the Mori–Tanaka method. Again, it is to be noted that the researchers presented the effect of defects on the engineering coefficient only.

Numerous works are presented on UD composites to investigate the effect of voids on elastic properties [12,17–20,22–25,28,29]. To the best of the authors' knowledge, the perturbation range of all EMPs and their distributions have not been comprehensively quantified for UD composite materials with a combination of different void shapes, sizes, and locations. In this research, we design a series of simulations to determine the local perturbation range of material properties of UD composite materials in the presence of different types of void contents, shapes, sizes, and locations. The local perturbation range of material properties of damaged UD composites can be used to calculate the strength

of the early-stage damage of the composite plates with multiple holes. In our Effects of Defects part II, the effect of microscale defects in the vicinity of holes is studied to predict the ultimate strength of composite plates with multiple open-holes and different fiber directions. The present paper is arranged in the following sequence. The experimental study and design of simulation for void modeling in a unit cell are discussed. Next, how to implement the periodic boundary conditions (PBCs) is explained to obtain the EMPs in the presence of voids. Further, the calculated EMPs and their perturbation range due to the void content are discussed.

## 2. Experimental Observations and the Process of Void Modeling

In this study, the UD carbon fiber-reinforced plastic (CFRP) composite material is considered to measure the diameter range of micro voids in the UD composite. The UD composite considered is assumed to be transversely isotropic [30]. The fiber diameter was 7 μm, and the fiber volume fraction was 50%. Interfaces between the fiber and matrix were assumed to be perfectly bonded [31]. The material properties of the UD composite are listed in Table 1, provided by the vendor ACP composites Inc. In Table 1, $E_{fl}$, and $E_{fv}$ are the modulus of elasticity of fibers along and normal to fiber directions, respectively. $v_{fl}$ and $v_{fv}$ are Poisson's ratio of fibers along and normal to fiber directions, respectively. $E$ and $v$ are the elastic modulus and Poisson's ratio of the matrix, respectively.

**Table 1.** Material properties of the fiber and matrix.

| T 300 Carbon Fiber | | | | Epoxy | |
|---|---|---|---|---|---|
| $E_{fl}$ (GPa) | $E_{fv}$ (GPa) | $v_{fl}$ | $v_{fv}$ | $E$ | $v$ |
| 230 | 15 | 0.25 | 0.07 | 5 | 0.16 |

According to our previous work, 20 specimens in total were planned to measure the void diameter to design practical simulations. The dimensions of specimens used in the tests were chosen based on ASTM C1275-95, ASTM C1359-96, and ISO/DIS 15490, which are 4 mm (thickness), 75 mm (length), and 8 mm (width). The specimens were placed in a material testing system (MTS) to run a fatigue test with 300,000 cycles. Next, the specimens were scanned by a scanning acoustic microscope (SAM), and the images obtained were analyzed by means of image processing software installed on SAM. All SAM images were taken at room temperature with a 100 MHz acoustic transducer with a lower and higher frequency range of ~25 MHz to ~500 MHz, with a peak near ~100 MHz. More than ~456 voids were imaged and quantified. The results showed that the diameters of the voids were in the range of 1.3 μm to 2.8 μm. In addition to void diameters, void shapes were not fully spherical. Therefore, the void shapes in spherical or ellipsoid form were assumed to be used in the numerical models [1]. Figure 1 shows the experimental work procedure that was carried out to measure void diameters and determine void shapes for the design of simulations.

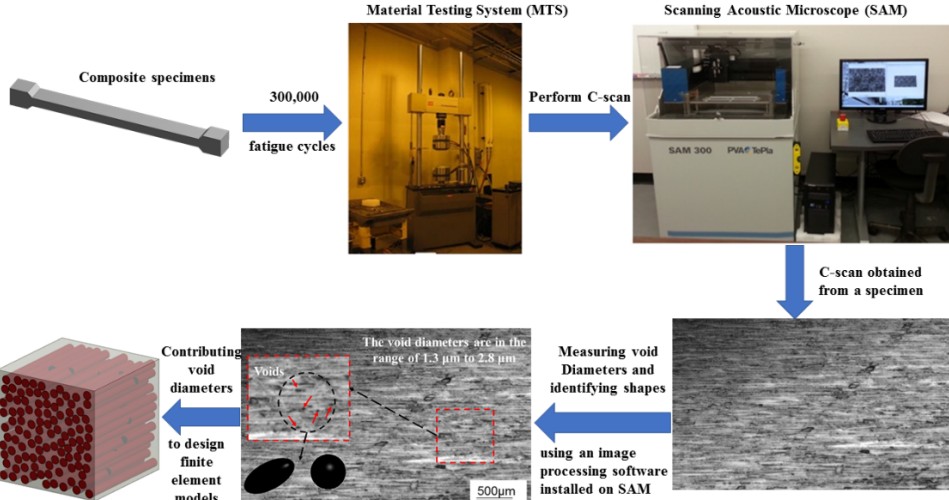

**Figure 1.** Experimental process to measure the range of void diameter for design simulation.

### 3. Design of Simulations

It was reported in the literature that in addition to void diameters, the void locations and void shapes may affect the degraded material properties [1,14,15,19,20]. In other words, the amount of degradation for each EMP may change due to different void shapes and void locations. Therefore, a comprehensive study is required to quantify not only the range of degraded material properties but also the uncertainty distribution of each EMP. Therefore, based on our experimental data, comprehensive simulations were designed using a unit cell concept. The unit cells, the micromechanical concepts, and the homogenization technique were employed to calculate the local degradation of composite materials due to the void content. To understand the perturbation of EMPs due to the voids [32–34], the unit cells were constructed with a 1%, 2%, 3%, and 5% void content in the unit cells for simulation. Figure 2 shows the design of simulations of the unit cells with a fixed void percentage. As shown in this figure, the simulations were designed for different configurations of the spherical, ellipsoid, and a combination of spherical and ellipsoid void contents. Please note that the ellipsoid void shapes are parallel to the fiber direction. The designed simulation was considered for 1%, 2%, 3%, and 5% void contents in the unit cells.

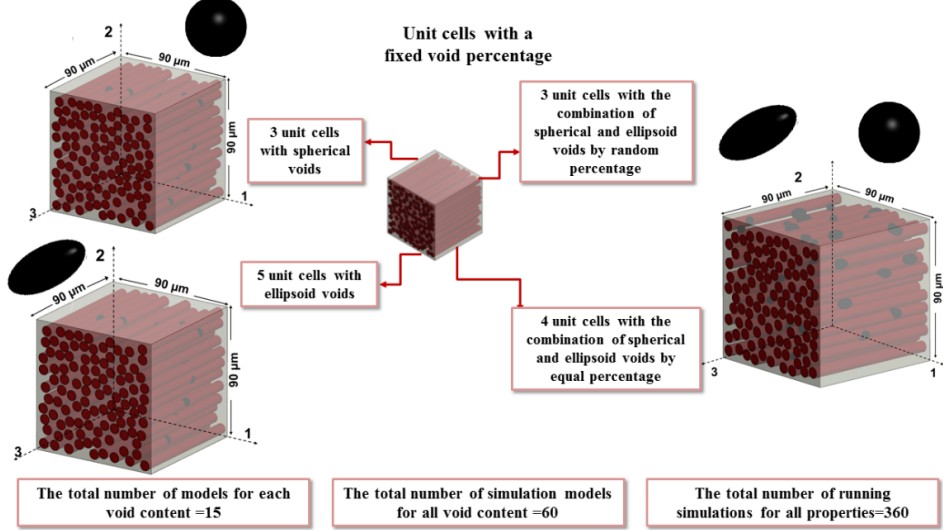

**Figure 2.** Design simulation for different configurations of the void content.

Sixty unit cells were considered to calculate the range of each EMP due to the voids. These unit cells were constructed as follows: 12 RVEs with spherical voids, 20 RVEs with

ellipsoid voids, 16 RVEs with a combination of spherical and ellipsoid voids in equal percentage, and 12 RVEs with a combination of spherical and ellipsoid voids with random different percentages. To thoroughly understand the statistical effect of void shapes, sizes, and locations on the constitutive coefficients, the EMPs were calculated for 60 unit cells with 1%, 2%, 3%, and 5% void percentages with 15 unit cells for each.

To determine the size of the unit cells with void contents, the procedure established in [21,22] was followed, and the size of the unit cell was calculated. The size of the unit cell for the UD composite with void contents was 90 μm. The material properties of the fiber and the matrix in our simulations are listed in Table 1. In this study, the unit cells in the pristine state and unit cells with voids were generated using a specific algorithm. In the pristine state cases, the location of fibers in the unit cell was determined using a random number generated from a normal distribution in such a way that the location of the fibers did not interfere with each other. Once unit cells were generated in the pristine state, the voids were placed in the RVE based on the same approach. The center and radii of the spherical and two axes of the ellipsoid voids were generated with random numbers generated from a normal distribution. The location and radius of the voids were constantly checked with the fiber locations, and the voids were placed in such a way that there was no contact/conflict between the fibers and the voids. This procedure was continued until the desired void content percentage was achieved.

## 4. Implementation of Periodic Boundary Conditions on Unit Cells to Quantify EMPs

Fiber-reinforced composite materials are assumed to have a repetitive structure herein. A unit cell is the smallest repeating component of microscopic composites that can represent an element containing their microscopic constituent [35]. Unit cells can be used to estimate material properties from the micro-scale to the meso-scale. Finding material properties for composites is very challenging (shear and thickness direction properties) and experimentally expensive. Therefore, the unit cell is used to quantify EMPs in composite materials. Figure 3a shows the periodicity of the composite and unit cell configuration with voids.

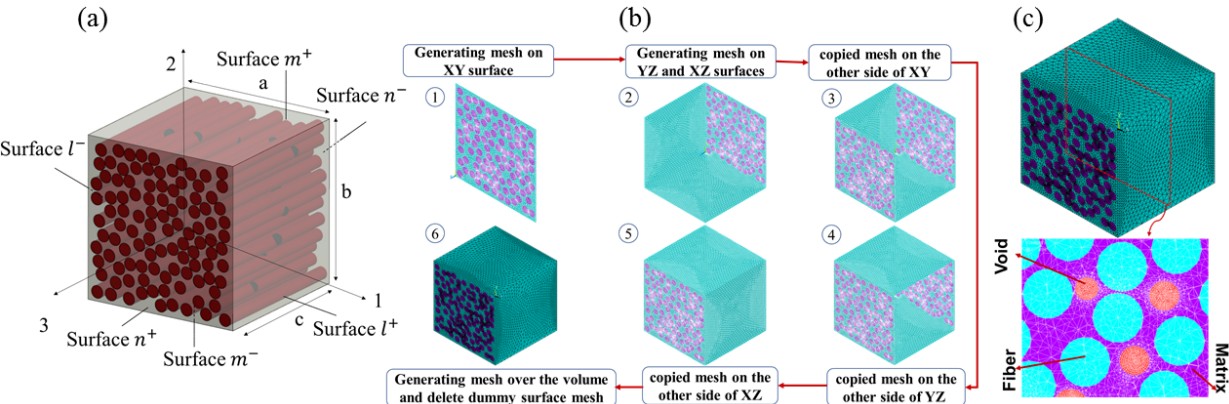

**Figure 3.** (**a**) Unit cell, (**b**) Process of generating an identical mesh on opposite surfaces, (**c**) Mesh generation ins a section of the unit cell with voids.

Since the UD composite can be considered to have a periodical structure, PBCs must be applied to the unit cells to find the EMPs of composite materials. In other words, unit cells that are located next to each other have the same mode deformation and they do not have any separation or overlap between each other. The PBCs are shown in Equation (1) [35].

$$u_i = \overline{S}_{ij}\, x_j + v_i \quad i = 1, 2, 3 \tag{1}$$

where $u_i$ represents displacement, $x_j$ represents the location, and $\overline{S}_{ij}$ represents the average strains. $v_i$ is the periodic part of displacement on the boundary conditions, which are un-

known and are related to the global load. If Equation (1) is written for two opposite surfaces $P^+$ and $P^-$ and their results are subtracted from each other, the PBCs can be presented by Equation (2), where $P^+$ and $P^-$ are $l^+$, $m^+$, $n^+$ and $l^-$, $m^-$, $n^-$ as demonstrated in Figure 3a.

$$u_i^{P+} - u_i^{P-} = \overline{S}_{ij} \left( x_i^{P+} - x_i^{P-} \right) \tag{2}$$

After implementing the PBCs, the average strains and the average stresses and EMPs are calculated. The average strain and average stress can be evaluated by [35].

$$\overline{\varepsilon}_{ij} = \frac{1}{V} \int_V \varepsilon_{ij} \, dV; \quad \overline{\sigma}_{ij} = \frac{1}{V} \int_V \sigma_{ij} \, dV; \quad C_{ij} = \frac{\overline{\sigma}_{ij}}{\overline{\varepsilon}_{ij}} \tag{3}$$

where $V$ is the volume of the periodic unit cell, $\overline{\varepsilon}_{ij}$ represents average strains, and $\overline{\sigma}_{ij}$ represents average stresses. It is important to emphasize that $\overline{S}_{ij}$ in Equation (2) is an arbitrary value. $\overline{S}_{ij}$ is considered to be 1 in this study.

Required PBCs are listed in Table 2 to calculate each constitutive coefficient [35]. To prevent rigid body motion, $u_x$, $u_y$, and $u_z$ are imposed to be zero where x, y, and z are at their minimum values. The mesh generated on the opposite surfaces needs to be identical to apply the PBCs [35–39]. Figure 3b shows the process of identical mesh generation over the unit cell surfaces. To fulfill this requirement, pre-mesh was generated on the master surfaces of the unit cell using dummy 2D linear elements (eight nodes with six degrees of freedom—Shell 281). This process can be seen in Figure 3b starting from step 1 to step 2. As shown in steps 1 and 2, the dummy surface meshes were generated on the surfaces of XY, YZ, and XZ. Next, the meshes were copied over to the opposite surfaces that are presented in steps 3, 4, and 5. The mesh generation was performed using an ANSYS built-in free surface mesh. The mesh generation on the matrix and the fibers was performed based on the surface meshes with tetrahedral linear elements. The dummy 2D linear elements were deleted after the meshing was performed over the volume, which is depicted in step 6 of Figure 3b. Figure 3c depicts the mesh generation of matrix, voids, and the matrix in a section of the unit cell.

**Table 2.** Implementing PBCs for 3D unit cells.

| Constitutive Coefficients | Surfaces $X_{max}/X_{min}$ | | | Surfaces $Y_{max}/Y_{min}$ | | | Surfaces $Z_{max}/Z_{min}$ | | |
|---|---|---|---|---|---|---|---|---|---|
| $C_{11}, C_{21}, C_{31}$ | $\Delta u_x = a$ | $\Delta u_y = 0$ | $\Delta u_z = 0$ | $\Delta u_x = 0$ | $\Delta u_y = 0$ | $\Delta u_z = 0$ | $\Delta u_x = 0$ | $\Delta u_y = 0$ | $\Delta u_z = 0$ |
| $C_{12}, C_{22}, C_{32}$ | $\Delta u_x = 0$ | $\Delta u_y = 0$ | $\Delta u_z = 0$ | $\Delta u_x = 0$ | $\Delta u_y = b$ | $\Delta u_z = 0$ | $\Delta u_x = 0$ | $\Delta u_y = 0$ | $\Delta u_z = 0$ |
| $C_{13}, C_{23}, C_{33}$ | $\Delta u_x = 0$ | $\Delta u_y = 0$ | $\Delta u_z = 0$ | $\Delta u_x = 0$ | $\Delta u_y = 0$ | $\Delta u_z = 0$ | $\Delta u_x = 0$ | $\Delta u_y = 0$ | $\Delta u_z = c$ |
| $C_{44}$ | $\Delta u_x = 0$ | $\Delta u_y = 0$ | $\Delta u_z = 0$ | $\Delta u_x = 0$ | $\Delta u_y = 0$ | $\Delta u_z = 0$ | $\Delta u_x = 0$ | $\Delta u_y = c$ | $\Delta u_z = 0$ |
| $C_{55}$ | $\Delta u_x = 0$ | $\Delta u_y = 0$ | $\Delta u_z = 0$ | $\Delta u_x = 0$ | $\Delta u_y = 0$ | $\Delta u_z = 0$ | $\Delta u_x = c$ | $\Delta u_y = 0$ | $\Delta u_z = 0$ |
| $C_{66}$ | $\Delta u_x = 0$ | $\Delta u_y = 0$ | $\Delta u_z = 0$ | $\Delta u_x = b$ | $\Delta u_y = 0$ | $\Delta u_z = 0$ | $\Delta u_x = 0$ | $\Delta u_y = 0$ | $\Delta u_z = 0$ |

The PBCs were applied based on the concept presented in earlier literature [37–39]. Figure 4 indicates the direction of applied loading, and the PCBs were applied to the nodes to calculate $C_{11}$, $C_{21}$, and $C_{31}$. If the PBCs were applied to all nodes over the volume, over-constraint occurs. To avoid over-constrained conditions, PBCs should be assigned to each pair of nodes only once. To satisfy this requirement, all pairs of nodes were designated as the reference and the dependent nodes in the corners. The reference and the dependent lines were defined without the corner nodes, and the reference and the dependent surfaces were defined without the corner nodes and the lines. The categorization process was performed using a MATLAB code. Figure 4 depicts the category of the contributed pair of nodes with PBCs at the corners, on the opposite lines, and over the opposite surfaces. The reference and the dependent nodes in the corners, on the lines, and over the surfaces are shown in blue and red color, respectively.

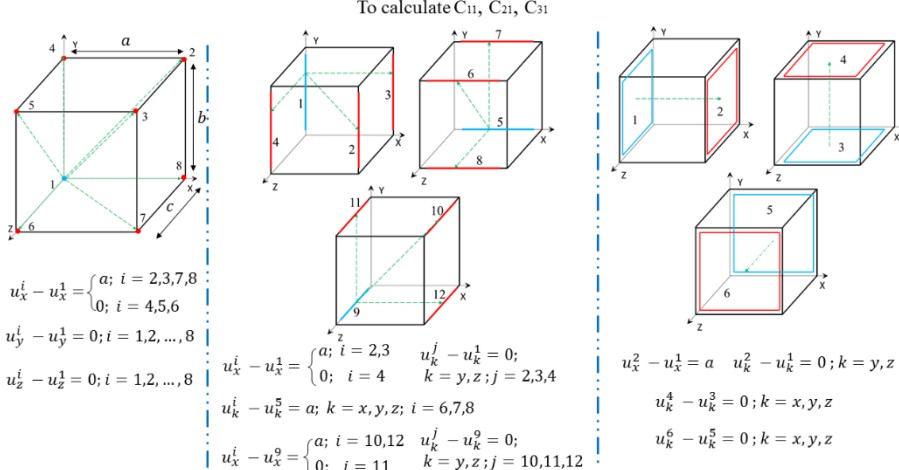

**Figure 4.** Categorized nodes, lines, and surface selection for applying PBCs.

The PBCs were generated using a MATLAB code in form of ANSYS interface language for desired constitutive coefficients. For further details, equations of the periodic boundary condition for calculating $C_{11}$, $C_{21}$, and $C_{31}$ are depicted in Figure 4. As shown in this figure, the external load was only applied to specific points (2, 3, 7, 8), lines (2, 3, 10, 12), and a surface (2) to quantify $C_{11}$, $C_{21}$, and $C_{31}$.

To implement the PBCs, ANSYS software based on equations in Figure 4 was employed. This result is in the range of linear elastic materials. To solve the problem, equations listed in Table 2 were rewritten in the form of an FEM for each desired constitutive coefficient as follows:

$$u_i^{P+} - u_i^{P-} = \overline{S}_{ij} \left( x_i^{P+} - x_i^{P-} \right) \ i,j, = 1,2,3 \ P = l,m,n$$
$$[K]g \ \{u\_i\} = \{F\} \tag{4}$$

where $\{F\}$ is equal to $\overline{S}_{ij} \left( x_i^{P+} - x_i^{P-} \right)$ and it can be a, b or c, which is 90 µm for each case, $\{u_i\}$ is the displacement, and $[K]^g$ is the coefficient of displacement. To find the EMPs after solving Equation (4), the stresses and the strains were extracted from each element. Then, based on Equation (3), the average stress and the average strain were calculated. The ANSYS interface was used to obtain integral stresses and strains for each constitutive coefficient.

## 5. Results and Discussion

First, the EMPs in the pristine state were calculated. This is required to accurately quantify the perturbation of EMPs when different void states are present in the microstructure. Linear tetrahedral elements (eight nodes with three degrees of freedom at each node—Solid 185) were employed to perform this study as mentioned earlier. To verify our new and faster approach and to calculate all of the coefficients with the 3D model (both element types and the procedure of implementation of PBCs on the unit cells), the results of Ref. [35] were replicated first, with both linear (Solid 185) and quadratic tetrahedral (20 nodes with three degrees of freedom at each node—Solid 187) elements. In Ref. [35], the EMPs of piezoelectric fiber composites, which are non-isotropic materials, were calculated. The results obtained by the quadratic tetrahedral elements in Ref. [35] were compared to results obtained by the linear tetrahedral elements in the present study. As shown in Figure 5, the EMPs, stress distribution, and strain distribution are presented and compared with each other. A comparison of the results showed that while being faster, the calculated EMPs, stress, and strain distributions using the linear tetrahedral element type had a good correlation with the existing literature [35]. It was observed that the computation time and memory required for the linear tetrahedral element type were lower (~20%) than the

simulation using the quadratic tetrahedral element type. Thus, in this study, all of the results were obtained using the linear tetrahedral elements.

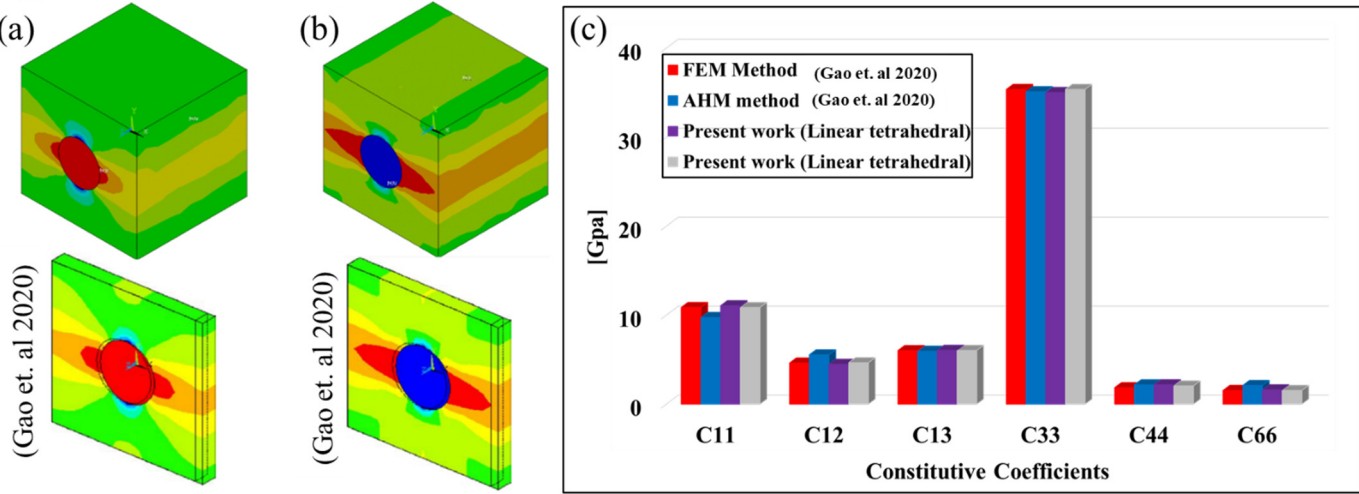

**Figure 5.** Verification of results with existing literature: (**a**) stress distribution, (**b**) strain distribution, and (**c**) EMPs.

Since the results obtained were verified with the results in the existing literature [26], the EMPs of the unit cell can be obtained with multiple fibers in the pristine state and the damaged state following the approach described above and the approach described below. To perform a convergence study, the number of elements on each surface of a unit cell was increased. The EMPs were calculated for each case. The results showed that the convergence requirement was met once each line of the unit cell and the circumferential line of the fiber was divided to 25 and 10 elements, respectively. Fifteen multi-fiber unit cell models in a pristine state were considered with different fiber locations to study the effect and quantify the variability of fiber location on EMPs. By applying PBCs and the constitutive matrix, the EMPs were obtained for the fiber-reinforced composite, which is listed in Table 2 using the method presented in the previous section. The locations of fibers were chosen using random numbers generated from the normal distribution. The mean of EMPs for 15 RVEs can be written due to different fiber locations as follows:

$$[C] = \begin{bmatrix} 8.46 & 1.27 & 2.06 & 0 & 0 & 0 \\ 1.27 & 8.46 & 2.06 & 0 & 0 & 0 \\ 2.06 & 2.06 & 116.46 & 0 & 0 & 0 \\ 0 & 0 & 0 & 4.88 & 0 & 0 \\ 0 & 0 & 0 & 0 & 4.87 & 0 \\ 0 & 0 & 0 & 0 & 0 & 3.54 \end{bmatrix} \ GPa \tag{5}$$

As shown in Equation (5), $C_{33}$ represents the EMP for the UD composite along the fiber direction. Directions 1 and 2 are perpendicular to the fiber direction. The EMP along the fiber direction is 116.46 GPa, $C_{11} = C_{22}$ is 1.27 GPa, and $C_{44} \approx C_{55}$, which was expected from the transverse isotropic properties of the UD composite. The distributions of EMPs in the pristine state due to different fiber locations are depicted in Figure 6. Figure 6a shows the bar plot and corresponding normal distribution of the actual material properties in GPa, whereas Figure 6b shows the box plot of each normalized coefficient. To the best of the authors' knowledge, the distribution of EMPs was not determined before using the proposed approach. Therefore, as a first guess, the normal distribution was chosen to demonstrate the EMP distributions of the unit cell in the pristine state. It can be seen that the $C_{44}$ and $C_{55}$ coefficients were most affected due to the fiber randomness. However, the variability of any coefficient was below 5%. $C_{33}$, which represents the material property

along the fiber direction, was least affected, and this can be considered another verification of the faster model adopted herein.

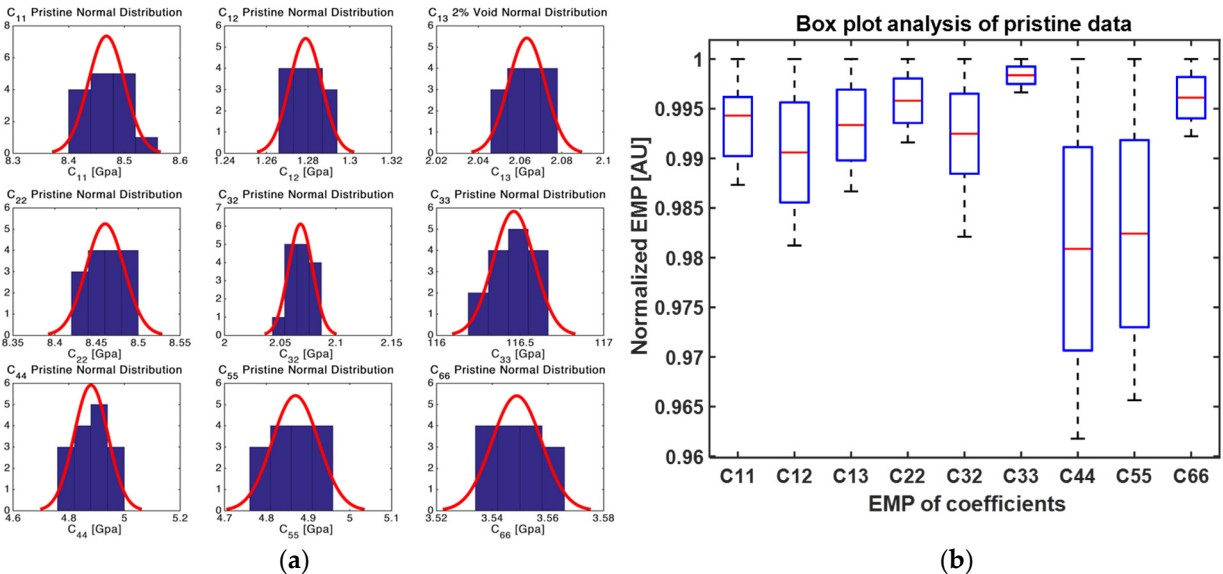

**Figure 6.** Distribution of EMPs in the pristine state, (**a**) normal distribution of constitutive coefficients (**b**) box plot showing, minimum first quartile, median, third quartile and maximum values of each constitutive constants.

### 5.1. Perturbation Range and Distributions of EMPs in the Presence of Different Percentages of Voids

In total, 360 simulations were performed to capture the perturbation range and distributions of EMPs of the UD composites in the presence of voids. In these unit cells, different void sizes, shapes, and locations were considered. The simulations were carried out for the UD composite with a 1%, 2%, 3%, and 5% void content. At a fixed void content, 15 unit cells with different void configurations were analyzed. The full matrix of EMPs and their distributions were calculated using the same process described in this article. Figure 7 presents the perturbation range of each EMP (the minimum, mean, median, and maximum values of EMPs) at different void percentages with a band of one standard deviation markdown. The figure shows that the EMPs or each coefficient of the constitutive matrix decreased with increasing void percentage in the unit cells. Moreover, Figure 7 shows the distributions of each EMP of the UD composites with a 3% void content. One standard deviation band of $C_{33}$ was narrower compared to the range of the bands of other EMPs since the void content had less effect on the EMPs along the fiber direction. A normal distribution was chosen to predict the distribution of the EMPs in the presence of 3% voids. This study showed that the normal distribution had a good correlation for predicting the distribution of EMPs in most cases. In Figure 7, box plots for each coefficient and the respective interquartile range for each coefficient are presented. The material property range of the interquartile range was identified on the material property scale in GPa. Standard deviations were also quantified and marked for all coefficients with a 3% void content.

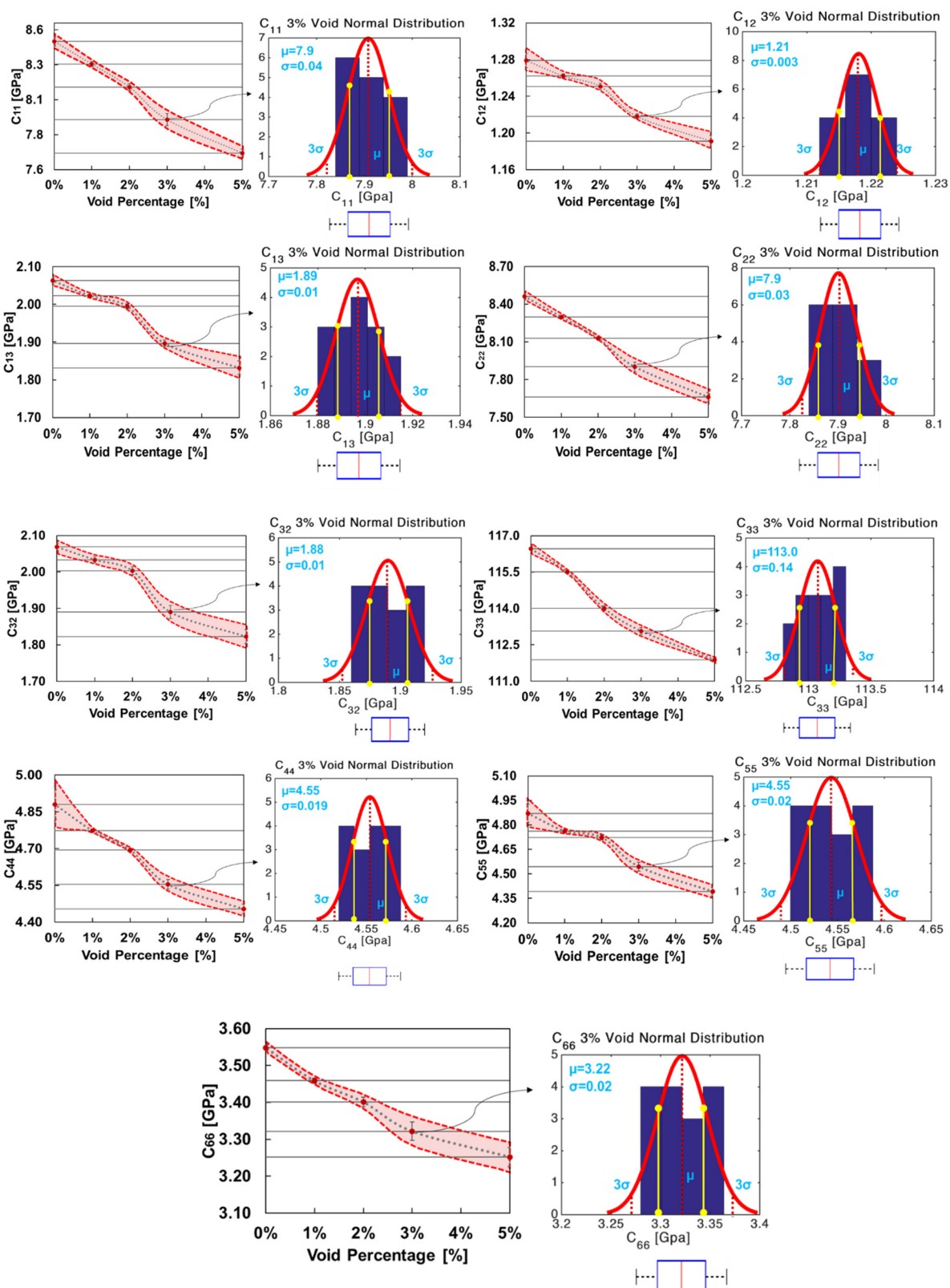

**Figure 7.** Perturbation range of EMPs due to increasing void content and distribution of EMPs of UD with a 3% void content.

As shown in Figure 7, the mean values of $C_{11}$, $C_{12}$, and $C_{13}$ were 7.9 GPa, 1.21 GPa, and 1.89 GPa, respectively. Additionally, the mean values of $C_{22}$, $C_{32}$, and $C_{33}$ were 7.9 GPa, 1.88 GPa, and 113 GPa, respectively. As illustrated in Figure 7, the mean values of $C_{44}$,

$C_{55}$, and $C_{66}$ for a 3% void content were 4.55 GPa, 4.55 GPa, and 3.22 GPa, respectively. The standard deviation of EMPs at a 3% void content was quantified using the normal distribution formula. It was found that the perturbations of EMPs in the presence of voids did not change linearly. Moreover, the EMPs did not change with the same percentage for each coefficient. For closer observation, the variability of the material constants in the pristine state in the presence of voids was normalized. The box plots of all the normalized material coefficients for 1%, 2%, 3%, and 5% void contents are presented in Figure 8 beside the pristine box plots. There are a few conclusions we can make from Figure 8. Overall, the $C_{13}$ and $C_{32}$ coefficients were more affected due to the presence of voids. At a higher percentage of voids, they were more affected. Due to the presence of voids, the pristine variability of $C_{44}$ and $C_{55}$ was significantly reduced, which is also evident from Figure 7. As the percentage of void increased, the randomness in the material coefficients increased. Overall, all material coefficients decreased with increasing void content but not with the same percentage.

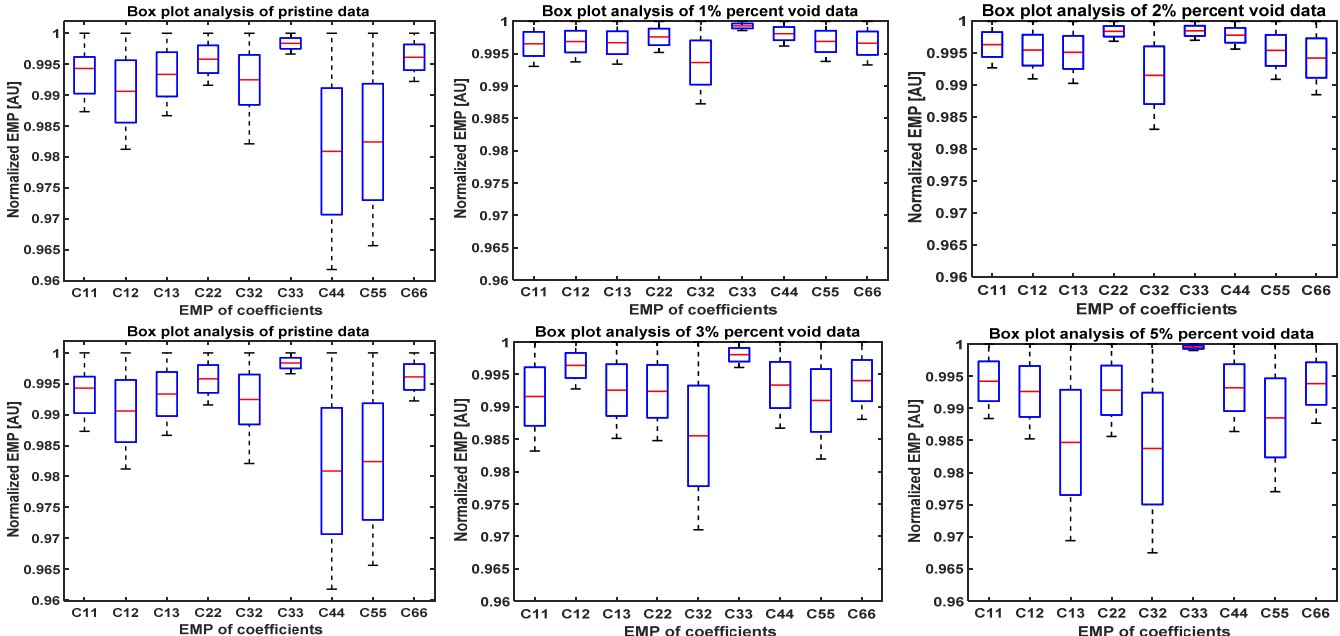

**Figure 8.** Distribution of EMPs with 1% and 2% void contents.

To present the quantitative degradation, the perturbation ranges of EMPs of the UD composites with 1% and 2% are listed in Equation (6). The results showed that $C_{66}$ degraded $-2.2\% \sim -2.86\%$ from its pristine state. It is concluded from the results that a lower void content (1% and 2%) had a significant effect on $C_{66}$ (associated with $\sigma_{xy} = C_{66}\,\varepsilon_{xy}$) and a lower effect on $C_{33}$ (associated with $\sigma_z = C_{33}\,\varepsilon_z$).

$$[C]^{1\%\ void} = \begin{bmatrix} -1.57 \sim -2.26\% & sym & sym & 0 & 0 & 0 \\ -1.00 \sim -1.62\% & -1.68 \sim 2.15\% & sym & 0 & 0 & 0 \\ -1.62 \sim 2.28\% & -1.05 \sim 2.31\% & -0.75 \sim 0.89\% & 0 & 0 & 0 \\ 0 & 0 & 0 & -2.01 \sim -2.39\% & 0 & 0 \\ 0 & 0 & 0 & 0 & -1.91 \sim 2.52\% & 0 \\ 0 & 0 & 0 & 0 & 0 & -2.20 \sim 2.86\% \end{bmatrix}$$

$$[C]^{2\%\ void} = \begin{bmatrix} -3.43 \sim -4.27\% & sym & sym & 0 & 0 & 0 \\ -1.76 \sim -2.64\% & -3.76 \sim -4.06\% & sym & 0 & 0 & 0 \\ -2.83 \sim -3.78\% & -2.33 \sim 3.98\% & -1.96 \sim 2.25\% & 0 & 0 & 0 \\ 0 & 0 & 0 & -3.06 \sim 4.02\% & 0 & 0 \\ 0 & 0 & 0 & 0 & -2.57 \sim 3.45\% & 0 \\ 0 & 0 & 0 & 0 & 0 & -3.60 \sim 4.70\% \end{bmatrix} \quad (6)$$

As listed in Equation (6), all of the coefficients in the matrix of the EMPs decreased when the void content was 1% and 2%. $C_{11}$ and $C_{22}$ decreased by approximately ~2%, and $C_{33}$ decreased by ~0.8% at a 1% void content. For the 2% void content, $C_{11}$ and $C_{22}$ decreased by approximately ~3.5%, and $C_{33}$ decreased by ~2%. $C_{11}$ and $C_{22}$ have more degradation compared to $C_{33}$ (fiber direction) since the matrix contributed to carry the load. All of the off-diagonal coefficients were also degraded due to the presence of voids. $C_{31}$ and $C_{32}$ decreased by ~2% and $C_{21}$ decreased by ~1% at a 1% void content. For the UD composites with a 2% void content, $C_{31}$ and $C_{32}$ decreased by ~3.5%, and $C_{21}$ decreased by ~2.5%. $C_{31}$ and $C_{32}$ had higher degradation compared to $C_{21}$ since the matrix carried less stress perpendicular to the fiber direction compared to the pristine state. $C_{44}$, $C_{55}$, and $C_{66}$ were degraded by almost ~2% and 3% for the unit cell with 1% and 2% void content, respectively. However, the degradation of $C_{66}$ was higher than that of $C_{55}$ and $C_{66}$ since the matrix contributed to carry the load.

The change in the percentage of mean values of each effective material property for the RVEs with a 5% void is written below. The normal distribution and box plot showing minimum first quartile, median, third quartile and maximum values of each constitutive constants are presented in Figure 9.

$$[C]^{5\%void} = \begin{bmatrix} -8.91 \sim -9.96\% & sym & sym & 0 & 0 & 0 \\ -6.16 \sim -7.54\% & -8.79 \sim -10.10\% & sym & 0 & 0 & 0 \\ -9.84 \sim -12.60\% & -10.41 \sim -13.32\% & -3.88\% \sim -3.97\% & 0 & 0 & 0 \\ 0 & 0 & 0 & -8.11\% \sim -9.36\% & 0 & 0 \\ 0 & 0 & 0 & 0 & -8.97\% \sim -10.67\% & 0 \\ 0 & 0 & 0 & 0 & 0 & -7.24\% \sim -9.48\% \end{bmatrix} \quad (7)$$

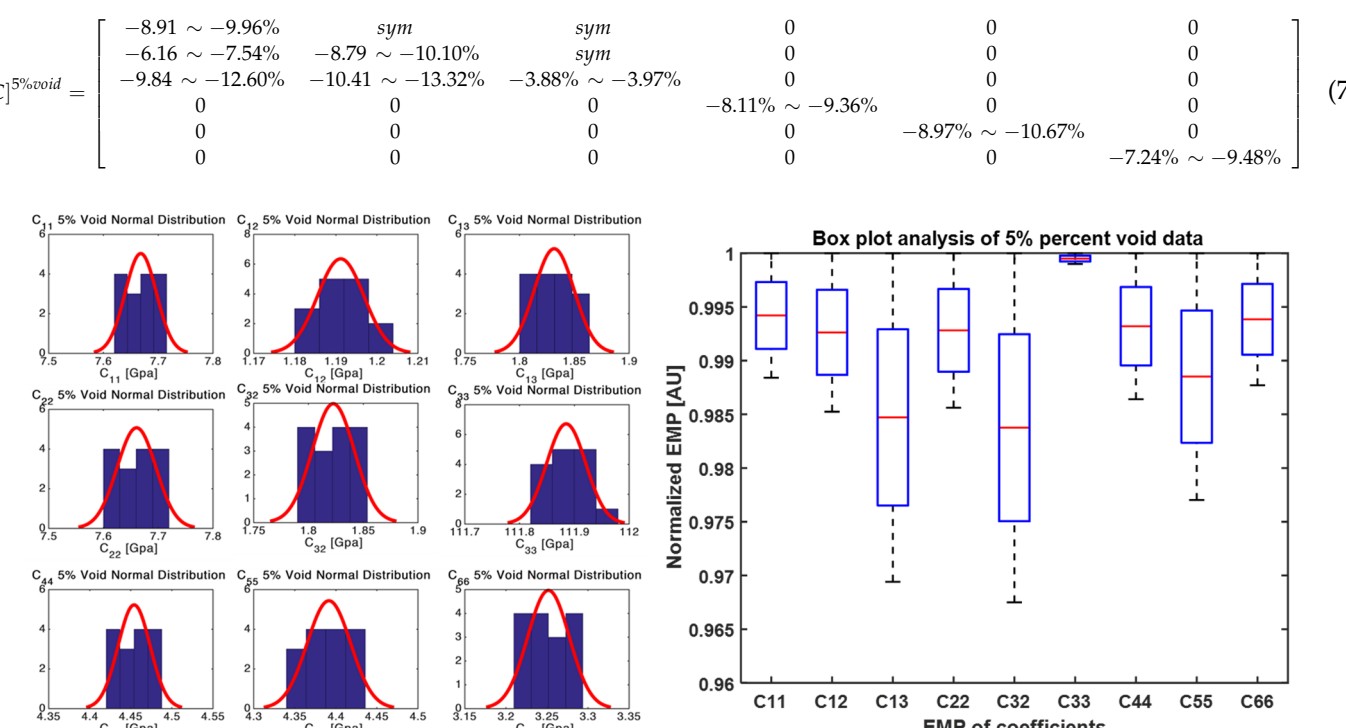

**Figure 9.** Distribution of EMPs with a 5% void content.

As listed in Equation (7), all of the coefficients in the matrix of the effective material property decreased once the void content was 5%. $C_{11}$ and $C_{22}$ decreased by approximately ~9% and 10%, respectively, and $C_{33}$ decreased by ~4%. All of the off-diagonal coefficients were also degraded due to the presence of voids. $C_{31}$ and $C_{32}$ were decreased by ~12% and 13%. $C_{21}$ was decreased by ~7%. $C_{44}$, $C_{55}$, and $C_{66}$ were decreased by almost ~10%.

### 5.2. Perturbation Range of the Engineering Constant in the Presence of Voids

The prediction of effective engineering constants was evaluated for the UD composite in Ref. [40]. The same procedure was followed to quantify the engineering constants herein. After obtaining the constitutive material coefficients and their respective distributions, the following expressions were used to calculate the engineering constants and their variability due to the presence of voids.

$$E_{33}^{eff} = C_{33}^{eff} - \frac{2\left(C_{13}^{eff}\right)^2}{C_{11}^{eff}+C_{12}^{eff}}$$

$$E_{11}^{eff} = C_{11}^{eff} - \frac{\left(C_{23}^{eff}\right)^2\left(C_{11}^{eff}-2\,C_{12}^{eff}\right)+C_{33}^{eff}\left(C_{12}^{eff}\right)^2}{\left(C_{23}^{eff}\right)^2-C_{33}^{eff}\,C_{11}^{eff}}$$

$$G_{23}^{eff} = C_{44}^{eff}$$

$$G_{12}^{eff} = \tfrac{1}{2}\left(C_{11}^{eff} - C_{12}^{eff}\right)$$

$$K_{33}^{eff} = \tfrac{1}{2}\left(C_{11}^{eff} + C_{12}^{eff}\right)$$

$$(8)$$

Figure 10 shows the effect of the void content on the effective engineering constants. As shown in Figure 10, the transverse elastic modulus ($E_{22}$) and in-plane shear modulus ($G_{12}$) had good correlations with the literature [34]. In this study, additional information on the effective engineering constant was quantified using a 3D RVE model. The effective elastic properties along the fiber directions ($E_{33}$) and out-plane shear modulus ($G_{23}$) and bulk modulus ($K_{33}$) decreased due to voids.

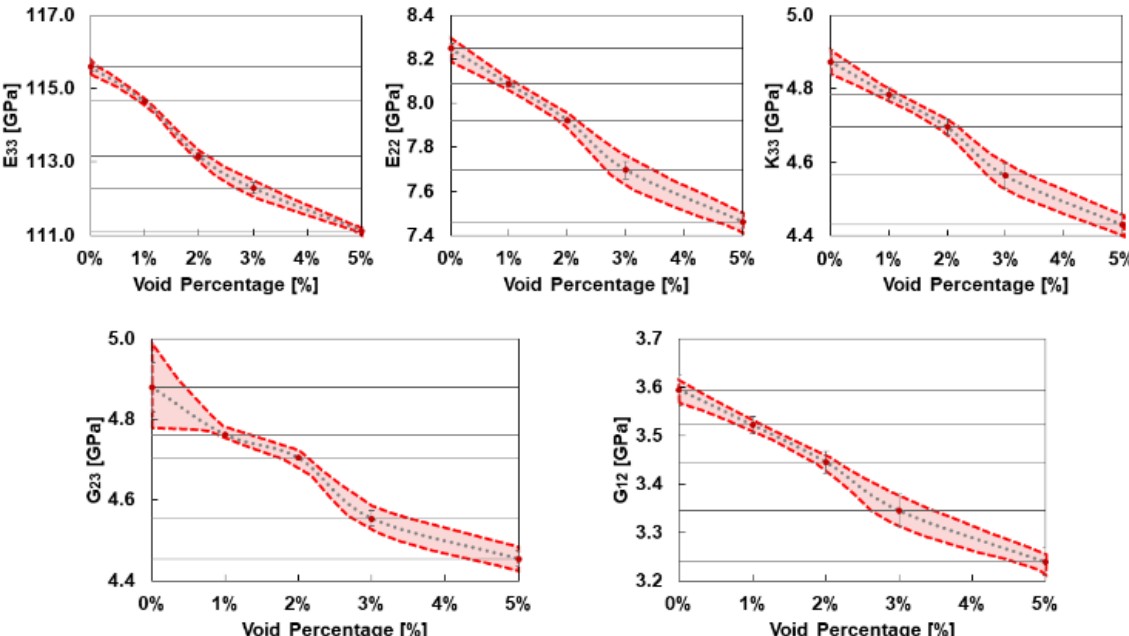

**Figure 10.** Engineering constant perturbation due to the voids.

## 6. Conclusions

In this study, the perturbations of EMPs as effects of microvoids were quantified in UD composite material. The experimental work was carried out using a scanning acoustic microscope (SAM) to measure the size and shape of the voids. Based on experimental data, the simulations were designed to consider the effect of all void parameters on the EMPs. Unit cells with 1%, 2%, 3%, and 5% void contents were introduced. The PBCs were applied to the unit cells to calculate the perturbation range and distribution of the EMPs due to different void contents and parameters. The results showed that the distribution of the degraded EMPs followed the normal distribution in most cases. The results confirmed that at a low percentage of voids (1% void content) the EMPs degraded and the amount of degradation of each EMP was not the same or linear. The void content had less effect on $C_{33}$ compared to $C_{11}$ and $C_{22}$ since the fibers contributed to carry the load. Based on the percentage of the degradation matrix, in Effect of Defect Part II we calculated the ultimate strength of the composite specimens with holes and degraded material properties around the hole. Together, Effect of Defect Part I and Part II present the multi-scale computational framework for determining the ultimate strength and perturbation of the

ultimate strength due to the degraded material property matrix if effectively used in progressive failure models.

**Author Contributions:** Conceptualization, S.B. and V.T.; methodology, V.T.; software, V.T.; validation, V.T., S.B.; formal analysis, V.T.; investigation, V.T.; resources, S.B.; data curation, S.B. and V.T.; writing—original draft preparation, V.T.; writing—review and editing, S.B.; visualization, V.T. and S.B.; supervision, S.B.; project administration, S.B.; funding acquisition, S.B. All authors have read and agreed to the published version of the manuscript.

**Funding:** This research was funded by NASA Langley Research Center, grant number NNL15AA16C.

**Institutional Review Board Statement:** Not applicable.

**Informed Consent Statement:** Not applicable.

**Data Availability Statement:** Data obtained from numerical experiments and simulations are available through integrated Material Assessment and Predictive Simulation Laboratory (i-MAPS) website upon request. Please contact the corresponding offer.

**Acknowledgments:** The author would like to acknowledge the NASA Langley Research Center (LaRC) for funding the research under Contract No. NNL15AA16C. The authors would also like to thank H. Berger at the University of Magdeburg for valuable discussions on the FEM simulation of RVE.

**Conflicts of Interest:** The authors declare no conflict of interest.

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
