# Peer review of "Effect of Defects Part I: Degradation of Constitutive Coefficients as an Input to the Composite Failure Model with Microvoids and Porosity"

_jcs, doi:10.3390/jcs6020037_

Round 1

Reviewer 1 Report

This research reports on an investigation into the effect of mechanical defects on the mechanical failure of composite materials containing micro-voids and porosity in which the appropriate material properties to a composite progressive failure model has been provided and a guideline for using an appropriate quantified perturbation associated with each coefficient has been presented. 

This is an interesting research with tangible results which can potentially be published in this journal. However, I don't recommend the manuscript's publication in its current form as Major revision is required with an interest in improving this research quality. My comments are as follows:

1) It's unclear and ambiguous as to how the experimental tests associated with the measurements of voids diameter are conducted. Fig.1 must be enhanced by providing more details of the experimental apparatus and test procedures. 

2) The reviewer has difficulties understanding how the experimental parameters were utilized to assist in the micromechanics-based finite element method. More clarification is required.

3) Have authors experimentally acquired the material properties of fiber and matrix? if not, it must be backed up by an appropriate reference. 

4) Can authors comment on the application of the proposed method to a perfectly fabricated composite structure free of voids or composite structures under various boundary and environmental conditions?

5) The literature review  must be enhanced by incorporating some of the latest published benchmarks in the area of composite failures under diverse loading conditions. As such, authors must discuss the following papers in the introduction portion of their manuscript. 

"Investigating the fatigue and mechanical behaviour of 3D printed woven and nonwoven continuous carbon fibre reinforced polymer (CFRP) composites", Composites Part B: Engineering, Volume 212, 1 May 2021, 108704      

"Localized failure analysis of internally pressurized laminated ellipsoidal woven GFRP composite domes: Analytical, numerical, and experimental studies", Archives of Civil and Mechanical Engineering, Vol:19, pp:1235-1250.   
6) The description of FEA modeling requires more detailed explanations. For instance:

-what kind of element has been used for FE simulation?

-How have boundary conditions been prescribed to the FE model analysis?

-Have you conducted a convergence study for results accuracy and reliability?

If the above comments are carefully addressed in the revised manuscript, it can then be considered for publication.

Author Response

Please find the attached file with our responses 

Reviewer 2 Report

The manuscript "Effect of Defects Part I: Degradation of Composite Constitutive Coefficients due to Microvoids and Porosity" is a of a good quality. The results are presented in a structured and qualitative way. The quality of figures and statistics particularly plausible. However, before the work can be published, there are a few drawbacks that should be addressed:

* What kind of epoxy was used? How was the modulus measured? It is strain rate and temperature dependent -- indicate at what parameters it was obtained

*Nothing is said about the effect of voids on ageing and diffusion. This is of high importance for long-term performance. For example see works by Starkova et al. or Gagani et al.

*The stages of possible damage are not described sufficiently. see recent work by Krauklis et al.: Should be four: (1) manufacturing of components/fibers/prepregs; (2) Composite manufacturing (3) Installation; (4) Exploitation/ageing + recycling [Krauklis, A.E.; Karl, C.W.; Gagani, A.I.; Jørgensen, J.K. Composite Material Recycling Technology—State-of-the-Art and Sustainable Development for the 2020s. J. Compos. Sci. 2021, 5, 28. https://doi.org/10.3390/jcs5010028]

*Check the manuscript for typos -- commented article attached for your convenience.

//Minor revision

Author Response

Please find the attached file with the responses 

Round 2

Reviewer 1 Report

The authors have satisfactorily revised the manuscript. This version is acceptable for publication.